A new testudinoid turtle from the middle to late Eocene of Vietnam

Garbin Rafaella C. 1 rafaella.decarvalhogarbin@unifr.ch
http://orcid.org/0000-0003-2100-6164 Böhme Madelaine 2 3
Joyce Walter G. 1
1 Department of Geosciences, University of Fribourg , Fribourg , Switzerland
2 Department of Geosciences, Eberhard-Karls-Universität Tübingen , Tübingen , Germany
3 Senckenberg Center for Human Evolution and Palaeoecology, Eberhard-Karls-Universität Tübingen , Tübingen , Germany
Sues Hans-Dieter
Electronic publication date: 2019 Feb 18
Publication date: 2019
Volume: 7
Electronic Location ID: e6280
Received 2018 Sep 7; Accepted 2018 Dec 12
Copyright: © 2019 Garbin et al.
Copyright year: 2019
Copyright holder: Garbin et al.
License: This is an open access article distributed under the terms of the Creative Commons Attribution License, which permits unrestricted use, distribution, reproduction and adaptation in any medium and for any purpose provided that it is properly attributed. For attribution, the original author(s), title, publication source (PeerJ) and either DOI or URL of the article must be cited.
License URL: https://creativecommons.org/licenses/by/4.0/

Keywords: Paleontology, Geoemydidae, Polymorphism, Intraspecific variation, Testudinoidea

Funding: Swiss National Science Foundation 20021_153502/1, 2 German Science Foundation BO 1550/11-1, 2 This work is funded by Swiss National Science Foundation 20021_153502/1, 2 and excavations in Na Duong have been supported by the German Science Foundation (BO 1550/11-1, 2). The funders had no role in study design, data collection and analysis, decision to publish, or preparation of the manuscript.

==============================
Background

Testudinoidea is a major clade of turtles that has colonized different ecological environments across the globe throughout the Tertiary. Aquatic testudinoids have a particularly rich fossil record in the Tertiary of the northern hemisphere, but little is known about the evolutionary history of the group, as the phylogenetic relationships of most fossils have not been established with confidence, in part due to high levels of homoplasy and polymorphism.

Methods

We here focus on describing a sample of 30 testudinoid shells, belonging to a single population that was collected from lake sediments from the middle to late Eocene (35–39 Ma) Na Duong Formation in Vietnam. The phylogenetic placement of this new material is investigated by integrating it and 11 other species of putative geoemydids from the Eocene and Oligocene to a recently published matrix of geoemydid turtles, that embraces the use of polymorphic characters, and then running a total-evidence analysis.

Results

The new material is highly polymorphic, but can be inferred with confidence to be a new taxon, Banhxeochelys trani gen. et sp. nov. It shares morphological similarities with other southeastern Asian testudinoids, Isometremys lacuna and Guangdongemys pingi, but is placed phylogenetically at the base of Pan-Testuguria when fossils are included in the analysis, or as a stem geoemydid when other fossils are deactivated from the matrix. The vast majority of other putative fossil geoemydids are placed at the base of Pan-Testuguria as well.

Discussion

The phylogenetic placement of fossil testudinoids used in the analysis is discussed individually and each species compared to Banhxeochelys trani gen. et sp. nov. The high levels of polymorphism observed in the new taxon is discussed in terms of ontogenetic and random variability. This is the first time that a large sample of fossil testudinoids has its morphological variation described in detail.

Introduction

Testudinoids (Cryptodira, Testudines) are an ecologically diverse and speciose clade of turtles (Ernst & Barbour, 1989) that colonized many terrestrial and freshwater environments over the course of the Tertiary (Sukhanov, 2000; Lapparent De Broin, 2001; Danilov, 2005; Vlachos, 2018) and now have a near global distribution with 190 extant species (Turtle Taxonomy Working Group (TTWG), 2017). Geoemydids, one of the four primary clades of Testudinoidea, have a particularly rich fossil record in the Paleogene of the northern hemisphere (Claude & Tong, 2004; Hervet, 2004a; Hutchison, 2006; Vlachos, 2018), especially in freshwater environments. Notable forms from this time interval include the North American Echmatemys lineage (Hay, 1908; Vlachos, 2018) and the European Palaeoemys and Ptychogaster lineages (Danilov, 2005; Hervet, 2006). The Paleogene record is still rather poor for the group in Asia (Gilmore, 1931), but rich remains have recently been described from the early Oligocene of Thailand (i.e., Hardella siamensis and Mauremys thanhinensis; Claude, Suteethorn & Tong, 2007) and the late Eocene of China (i.e., Guangdongemys pingi and Isometremys lacuna; Claude et al., 2012).

Although some effort has recently been placed on resolving the morphology and taxonomy of Paleogene geoemydids (Hervet, 2004a; 2004b; Claude & Tong, 2004; Claude, Suteethorn & Tong, 2007; Claude et al., 2012), little is still known about the evolutionary history of the group, as the phylogenetic relationships of most fossils have not been established with confidence (Claude et al., 2012). This is possibly related to high levels of homoplasy and polymorphism found in geoemydids (and testudinoids, in general) that precludes the performance of traditional phylogenetic analysis (Joyce & Bell, 2004; Garbin, Ascarrunz & Joyce, 2018).

A sample of approximately 100 geoemydid skeletons, mostly incomplete shells, was recently collected from the middle to late Eocene Na Duong Formation as exposed at the Na Duong coal mine in Vietnam (Böhme et al., 2013). The purpose of this contribution is to describe this new geoemydid material, which is considered to represent a single, new species based on the 30 best preserved specimens, and to assess its phylogenetic significance by placing it in a character taxon matrix that embraces polymorphic characters.

Geological Setting

The available sample of approximately 100 specimens was collected between 2009 and 2012 from the Na Duong Formation at the Na Duong coal mine, which is located in Loc Binh district, Lạng Sơn Province, Vietnam (Fig. 1). The Na Duong Formation is a 240 m thick continental deposit consisting of marly claystones, siltstones, and fine to medium-grained sandstones intercalated with lignite seams and extractable coal seams (Böhme et al., 2011, 2013). The vast majority of vertebrate finds made at this locality originate from a single horizon at the base of the main lignite seam (layer 80 of Böhme et al., 2011). Based on sedimentology, mineralogy, fauna and flora, the lignitic shales of layer 80 are interpreted as representing a shallow, acidic lake deposit (Böhme et al., 2013). Testudinoids are by far the most abundant vertebrates from this horizon (see Böhme et al., 2013). The same horizon has otherwise yielded trionychid turtles, cyprinid and amiid fishes, three species of crocodiles, and anthracothere and rhinocerotid mammals (Böhme et al., 2013). As all testudinoid specimens described herein were collected from about 50 cm of the same stratigraphic horizon, they are thought to represent a true population. Magnetostratigraphy in combination with the anthracotherid and rhinocerotid mammals suggest a late middle to late Eocene age (i.e., upper Bartonian—late Priabonian) for the Na Duong formation. For a more detailed discussion, please refer to Böhme et al. (2013).

Figure 1 Map of Southeast Asia showing Paleogene localities that yielded geoemydid turtles.

Stars: Maoming basin, China (eate Eocene); Krabi basin, Thailand (late Eocene—early Oligocene); Na Duong basin, Vietnam (middle to late Eocene). Based on UN map No. 4365.

Nomenclatural Acts

The electronic version of this article in Portable Document Format will represent a published work according to the International Commission on Zoological Nomenclature (ICZN), and hence the new names contained in the electronic version are effectively published under that Code from the electronic edition alone. This published work and the nomenclatural acts it contains have been registered in ZooBank, the online registration system for the ICZN. The Zoobank Life Science Identifiers (LSIDs) can be resolved and the associated information viewed through any standard web browser by appending the LSID to the prefix http://zoobank.org/. The LSID for this publication is urn:lsid:zoobank.org:pub:D2620202–9814–4F81–BFA4–043EB1B6F948. The online version of this work is archived and available from the following digital repositories: PeerJ, PubMed Central and CLOCKSS.

Systematic Paleontology

TESTUDINES Batsch, 1788

CRYPTODIRA Cope, 1868

TESTUDINOIDEA Fitzinger, 1826

TESTUGURIA Joyce, Parham & Gauthier, 2004

PAN-GEOEMYDIDAE Joyce, Parham & Gauthier, 2004

Banhxeochelys trani gen. et sp. nov.

Holotype. GPIT/RE/09760 (Fig. 2), a nearly complete shell lacking left peripherals VII–XI, left costals VII and VIII, the pygal, part of right peripherals VI and VII, and a part of the left xiphiplastron.

Figure 2 GPIT/RE/09760, Banhxeochelys trani gen. et sp. nov., holotype, subadult, middle to late Eocene (Priabonian) of Vietnam.

(A) Photograph of carapace. (B) Illustration of carapace. (C) Photograph of plastron. (D) Illustration of plastron. Abbreviations: An, anal scute; co, costal; epi, epiplastron; Gu, gular scute; Hu, humeral scute; hyo, hyoplastron; Ma, marginal scute; mdf, musk duct foramen; ne, neural; nu, nuchal; per, peripheral; spy, suprapygal; Ve, vertebral scute; xi, xiphiplastron.

Type locality and horizon. Na Duong coal mine, Long Binh District, Lạng Sơn Province, Vietnam; base of the main lignite seam (layer 80 of Böhme et al., 2013), upper Bartonian–late Priabonian (34–39 Ma), late middle to late Eocene (Böhme et al., 2013).

Etymology. “Bánh xèo” is a dish from Vietnam that resembles a crepe or pancake. “Chelys” derived from the Greek, meaning turtle. The genus name is formed in allusion to the highly compressed, pancake-like preservation of the type material. The species name honours Đặng Ngọc Trần, retired director of the International Cooperation Division of the Department of Geology and Minerals of Vietnam, for his untiring support of the excavations carried out by teams from the University of Tübingen from 2009 to 2012.

Diagnosis. Banhxeochelys trani gen. et sp. nov. can be diagnosed as a representative of Testudinoidea by the presence of a contact between plastral and marginal scutes and the corresponding absence of inframarginal scutes II and III, as a representative of Pan-Testuguria by the presence of a pygal bone that is wider than long, as a representative of Testuguria by the presence of short anal scutes, and as a representative of Pan-Geoemydidae by the presence of anterior musk duct foramina (mdf; Fig. 2), complete intersection of the pygal bone by the intersulcus of marginals XII (Figs. 3 and 4), a median keel on the carapace, and a cervical scute (Figs. 2, 3 and 5).

Figure 3 GPIT/RE/09735, Banhxeochelys trani gen. et sp. nov., adult, middle to late Eocene of Vietnam.

(A) Photograph of carapace. (B) Illustration of carapace. (C) Photograph of plastron. (D) Illustration of plastron. Abbreviations: An, anal scute; co, costal; ent, entoplastron; epi, epiplastron; Gu, gular scute; Hu, humeral scute; hyo, hyoplastron; Ma, marginal scute; ne, neural; nu, nuchal; per, peripheral; spy, suprapygal; Ve, vertebral scute.

Figure 4 GPIT/RE/09747, Banhxeochelys trani gen. et sp. nov., subadult, middle to late Eocene of Vietnam.

(A) Photograph of carapace. (B) Illustration of carapace. (C) Photograph of plastron. (D) Illustration of plastron. Abbreviations: Ab, abdominal scute; An, anal scute; Ma, marginal scute; ne, neural; per, peripheral; Pl, pleural scute; py, pygal; Ve, vertebral scute; xi, xiphiplastron.

Figure 5 GPIT/RE/09733, Banhxeochelys trani gen. et sp. nov., adult, middle to late Eocene of Vietnam.

(A) Photograph of carapace. (B) Illustration of carapace. (C) Photograph of plastron. (D) Illustration of plastron. Abbreviations: Ab, abdominal scute; An, anal scute; Ce, cervical scute; co, costal; ent, entoplastron; epi, epiplastron; Gu, gular scute; Hu, humeral scute; ne, neural; nu, nuchal; per, peripheral; Pl, pleural scute; Ve, vertebral scute; xi, xiphiplastron.

The following combination of characters is unique to this taxon: neural IV octagonal, neural V square, the remaining neurals hexagonal with anterior short-sides, two lateral keels on the carapace in juveniles (Fig. 2), a broad trapezoidal cervical scute (Figs. 2–4), wide suprapygal II that almost reaches peripheral X (Fig. 4), entoplastron intersected anteriorly by gularohumeral sulcus (Figs. 2 and 3) and posteriorly by humeropectoral sulcus (Figs. 3 and 5), and short epiplastral lip in visceral view (Fig. 6).

Figure 6 GPIT/RE/09749, Banhxeochelys trani gen. et sp. nov., subadult, middle to late Eocene of Vietnam.

(A) Photograph of carapace. (B) Illustration of carapace. Abbreviations: Ce, cervical scute; co, costal; epi, epiplastron; Gu, gular scute; ne, neural; per, peripheral; Pl, pleural scute; Ve, vertebral scute.

Referred material. Although the type locality has yielded remains of more than 100 turtles, we here only describe and refer the 30 best preserved specimens, as the remaining fossils are too fragmentary or too poorly preserved to provide useful morphological data: GPIT/RE/09749 (Fig. 6), subadult, incomplete carapace and epiplastra; GPIT/RE/09731 (Fig. 7), adult, complete carapace and plastron; GPIT/RE/09732 (Fig. 8), adult, incomplete carapace and plastron; GPIT/RE/09733 (Fig. 5), adult, complete carapace and plastron; GPIT/RE/09739, juvenile, incomplete carapace and plastron (rear part missing); GPIT/RE/09750, juvenile, incomplete carapace and complete plastron; GPIT/RE/09751, adult, complete carapace and incomplete plastron; GPIT/RE/09752, adult, epiplastra, entoplastron, hyoplastra, and incomplete carapace; GPIT/RE/09753, subadult, complete carapace and plastron; GPIT/RE/09754, adult, almost complete carapace and plastron; GPIT/RE/09736, adult, incomplete plastron; GPIT/RE/09755, adult, incomplete carapace and plastron; GPIT/RE/09748, subadult, incomplete carapace and plastron; GPIT/RE/09741, juvenile, incomplete carapace and plastron; GPIT/RE/09756, adult, incomplete carapace and plastron (xiphiplastra missing); GPIT/RE/09742, juvenile, incomplete carapace and plastron (xiphiplastra missing); GPIT/RE/09757, subadult, almost complete carapace and plastron; GPIT/RE/09737, adult, incomplete carapace and plastron; GPIT/RE/09738 (Fig. 9), adult, almost complete and plastron; GPIT/RE/09758, adult, almost complete carapace and complete plastron; GPIT/RE/09759 (Fig. 10), juvenile, complete carapace and plastron; GPIT/RE/09760 (Fig. 2), subadult, almost complete carapace and plastron; GPIT/RE/09743 (Fig. 11), juvenile, almost complete carapace and plastron; GPIT/RE/09740, juvenile, incomplete carapace and plastron; GPIT/RE/09734, adult, incomplete carapace and almost complete plastron; GPIT/RE/09735 (Fig. 3), adult, almost complete carapace and complete plastron; GPIT/RE/09745, subadult, incomplete carapace and plastron; GPIT/RE/09746, subadult, incomplete carapace and plastron; GPIT/RE/09747 (Fig. 4), subadult, posterior rear of carapace, hypoplastra and xiphiplastra; GPIT/RE/09744, juvenile, incomplete carapace and plastron.

Figure 7 GPIT/RE/09731, Banhxeochelys trani gen. et sp. nov., adult, middle to late Eocene of Vietnam.

(A) Photograph of carapace. (B) Illustration of carapace. (C) Photograph of plastron. (D) Illustration of plastron. Abbreviations: Ab, abdominal scute; An, anal scute; Ce, cervical scute; co, costal; epi, epiplastron; Hu, humeral scute; hyo, hyoplastron; ne, neural; nu, nuchal; per, peripheral; Pl, pleural scute; py, pygal; xi, xiphiplastron.

Figure 8 GPIT/RE/09732, Banhxeochelys trani gen. et sp. nov., adult, middle to late Eocene of Vietnam.

(A) Photograph of carapace. (B) Illustration of carapace. (C) Photograph of plastron. (D) Illustration of plastron. Abbreviations: co, costal; epi, epiplastron; Hu, humeral scute; hypo, hypoplastron; ne, neural; nu, nuchal; Pl, pleural scute; Ve, vertebral scute; xi, xiphiplastron.

Figure 9 GPIT/RE/09738, Banhxeochelys trani gen. et sp. nov., adult, middle to late Eocene of Vietnam.

(A) Photograph of carapace. (B) Illustration of carapace. (C) Photograph of plastron. (D) Illustration of plastron. Abbreviations: co, costal; ent, entoplastron; epi, epiplastron; Fe, femoral scute; Gu, gular scute; Ma, marginal scute; ne, neural; nu, nuchal; Pe, pectoral scute; per, peripheral; Pl, pleural scute; Ve, vertebral scute; sp, suprapygal; xi, xiphiplastron.

Figure 10 GPIT/RE/09759, Banhxeochelys trani gen. et sp. nov., juvenile, middle to late Eocene of Vietnam.

(A) Photograph of carapace. (B) Illustration of carapace. (C) Photograph of plastron. (D) Illustration of plastron. Abbreviations: Hu, humeral scute; hyo, hyoplastron; Ma, marginal scute; ne, neural; nu, nuchal; per, peripheral; Pl, pleural scute; spy, suprapygal; xi, xiphiplastron.

Figure 11 GPIT/RE/09743, Banhxeochelys trani gen. et sp. nov., juvenile, middle to late Eocene of Vietnam.

(A) Photograph of carapace. (B) Illustration of carapace. (C) Photograph of plastron. (D) Illustration of plastron. Abbreviations: An, anal scute; co, costal; ent, entoplastron; epi, epiplastron; Hu, humeral scute; hyo, hyoplastron; ne, neural; nu, nuchal; per, peripheral; Ve, vertebral scute; xi, xiphiplastron.

Description

Preservation. The new turtle material from the Na Duong formation is characterized by strong dorsoventral compression caused by post-depositional deformation. This precludes making observations to the visceral sides of the carapace and the plastron in most specimens. The pattern of sulci that normally characterized the surface of turtles is furthermore missing in numerous specimens due to surficial weathering of the pyritized shells. Not a single individual of the new turtle is therefore known by a complete specimen that preserves all scute sulci and bone sutures. Nevertheless, the 30 best preserved individuals used herein (14 adults, eight subadults and eight juveniles) in combination, provide information regarding the majority of shell structures and intraspecific variation. The reaming specimens are too fragmentary or too poorly preserved to provide useful morphological data.

Size Classes. To better understand the implications of the morphological variation observed in our sample of Banhxeochelys trani, we classify specimens in three maturity categories (i.e., adult, subadult and juvenile) according to the median length of their hypoplastron (HypoML; Fig. 12). The median length of the hypoplastron was chosen as our criterion as this is the only bone that is intact in all specimens of interest. Specimens with a hypoplastron length of 40 mm or less are classified as juveniles, specimens with an average hypoML of approximately 60 mm as subadults, and specimens with a hypoML of 70 mm or more as adults.

Figure 12 Median length of the hypoplastron (HypoML) in a sample of 18 specimens of Banhxeochelys train.

The trend in the measurements show the presence of three size groups. Adults have a hypoplastron with 70 mm in length or more. Subadults have an average HypoML of 60 mm. Juveniles have a hypoplastron median length average of 40 mm.

Carapace. A median keel is present in animals of all size classes, with exception of GPIT/RE/09732. Two lateral keels, on the other hand, are only present in smaller specimens, here interpreted as juveniles or subadults (Fig. 2). The median keel is low and continuous, anteroposteriorly directed, and crosses the entire neural series anteroposteriorly (Figs. 2 and 10), starting over at neural I or II and sometimes reaching to first suprapygal (Figs. 3 and 4). In GPIT/RE/09749 (Fig. 6), the median keel starts at the nuchal, on the anterior region of vertebral I. In most specimens, the median keel spans from vertebral I to V, but is restricted in GPIT/RE/09748 and GPIT/RE/09742 to vertebrals II and III. The lateral keels are located over the costals, are closer to the neurals than the peripherals, and either extend anteroposteriorly from pleural scutes I to IV (from the posterior region of costal I to the anterior region of costal VII) or are restricted to pleurals II and III (between costals III to VI). Growth annuli are evident on the carapace of some specimens (e.g., GPIT/RE/09743 and GPIT/RE/09745), but no size trend is apparent.

Nuchal. The nuchal resembles that of other pan-geoemydids by being hexagonal, the anterior margin being wider than posterior margin, and its maximum width being located around mid-length. The median keel emerges at the posterior half of the nuchal in GPIT/RE/09749 (Fig. 6). The ventral side of the nuchal is exposed only in GPIT/RE/09751, but it is not possible to see any characteristics due to bad preservation.

Neurals. Eight neurals are present, with exception of GPIT/RE/09738, which shows nine neurals, probably due to an anomalous division of a neural VIII (Fig. 9). The neural bones vary in shape from anterior to posterior. Whereas neurals I to IV are longer than wide, neurals V to VIII are as wide as long or wider than long. Neural I is always squarish and either has rounded lateral margins that form an oval shape (in 57% of specimens), parallel lateral margins that form an overall rectangular shape (in 25% of specimens), or convergent lateral margins that form a triangular shape (in 18% of specimens). Neurals II to VIII are hexagonal with anterior short-sides, with the exception of neural IV, which is octagonal with short anterior and posterior sides, and of neural V, which is quadrangular or rounded. GPIT/RE/09759, a juvenile, and GPIT/RE/09748 are the only specimens that have all hexagonal neurals with short anterior sides, including neurals IV and V. Usually all neurals are crossed anteroposteriorly by the median keel, with exception of GPIT/RE/09748 and GPIT/RE/09742 that have a median keel restricted to neurals III to VI.

Costals. Eight pairs of costal bones are present in the carapace, which do not alternate in length as in testudinids. Costal I is the anteroposteriorly longest element, about twice as long as costal II. While costals II to VI have about the same length and width, costals VII and VIII are significantly small in both dimensions. Costals VII and VIII never contact their counterpart at midline, as they are always separated by the neural series. A contact of the left axillary buttress with the visceral side of costal I can be observed in GPIT/RE/09751. No other specimens show clear evidence for axillary or inguinal buttress or their contacts with the costals. The lateral contacts of the costals with the peripherals are described below.

Peripherals. Due to the strong dorsoventral compression of the material, the peripherals are poorly preserved. In the majority of the specimens, the peripherals are displaced or shifted ventrally to partially cover the costals (e.g., GPIT/RE/09732; Fig. 8). A total of 11 pairs of peripherals are present. In general, the peripherals are not serrated, with exception of posterior peripherals of some specimens (Fig. 4). The lateral peripherals do not form a gutter. While peripheral I contacts the nuchal bone medially, peripherals I–III contact costal I posteromedially. Peripheral IV contacts costal I and II and peripheral V contacts costals II and III medially. Peripheral VI contacts costal IV medially in GPIT/RE/09743, but costals IV and V in GPIT/RE/09759, the only two specimens where these contacts are visible. Peripheral VII contacts costals V and VI. Ventrally, peripherals IV to VI form the bridge by contacting the hyo- and hypo-plastron. In GPIT/RE/09743 it is possible to see the insertion of the axillary and inguinal buttresses at peripherals III and VII, respectively (Fig. 11). An anterior musk duct foramen can be observed on the ventral side of peripheral III in GPIT/RE/09751, GPIT/RE/09752, GPIT/RE/09758 and GPIT/RE/09760 (Fig. 2). The posterior musk duct foramen, on the other hand, is visible on the ventral side of peripheral VII in GPIT/RE/09751 and GPIT/RE/09758. These foramina are obscured in all other specimens. Peripherals VIII to XI can have small serrations on the lateral margin, as in GPIT/RE/09731 and GPIT/RE/09735 (Figs. 3 and 7). Peripheral VII contacts costal VI medially. Peripheral IX contacts only costal VII (e.g., GPIT/RE/09739) or costals IV and VII (GPIT/RE/09743). Peripheral X contacts costal VII and VIII medially or only costal VIII (e.g., GPIT/RE/09735 and GPIT/RE/09731). Although GPIT/RE/09747 exhibits a contact between peripheral X and the second suprapygal, this contact is absent in other specimens where this character is discernable. Peripheral XI contacts the pygal and the second suprapygal medially.

Suprapygals and pygal. There are two suprapygals in all specimens. Suprapygal I is small, as wide as long, and similar in size and shape to the last neural bones, by being squarish with parallel or convergent lateral sides. It contacts neural VIII anteriorly, the eighth costal bones laterally, and suprapygal II posteriorly in all specimens for which the contacts can be observed. Suprapygal II is larger than suprapygal I, at least two times wider than long, hexagonal, and crossed posteriorly by the posterior margin of vertebral V and by the midline sulcus of marginals XII. It contacts suprapygal I anteriorly, the eighth costal bones and 11th peripherals laterally, and the pygal, posteriorly. Although disarticulated, in GPIT/RE/09731 (Fig. 7) suprapygal II possibly contacts peripheral X laterally. The pygal bone is small, as wide as long, with parallel lateral sides, and has a median notch along the posterior margin, with exception of GPIT/RE/09735, which does not have a notch. As in most crown geoemydids, the pygal is completely intersected by the median sulcus formed by marginals XII. It contacts laterally both peripherals XI and suprapygal II, anteriorly.

Cervical scute. A cervical scute is present in all specimens where the anterior margin of the carapace is intact. The cervical is small, usually as wide as long (longer than wide in GPIT/RE/09738 and GPIT/RE/09744), and has anteriorly convergent lateral sides. In half of specimens (e.g., GPIT/RE/09735, GPIT/RE/09749 and GPIT/RE/09738), a notch is present along the posterior margin of the cervical. Some specimens (e.g., GPIT/RE/09749 and GPIT/RE/09741), by contrast, show a strong anterolateral constriction of the cervical scute (Fig. 6).

Vertebral scutes. There are five vertebral scutes in all specimens, with the exception of GPIT/RE/09738, which presents a small, anomalous scute between vertebrals IV and V (Fig. 9). Vertebral I is quadrate, with convergent or sinuous lateral sides, and has an anterior margin that is always wider than the posterior one. It can be longer than wide, or wider than long. The sulcus between pleural I and vertebral I always contacts the medial portions of marginal I. A small constriction (i.e., an anterolateral step) is present in the anterior region of this sulcus in GPIT/RE/09758 (not figured). Vertebral II is hexagonal, longer than wide, and has lateral sides with equal lengths. The anterior margin crosses neural I and the posterior margin neural III in all specimens. Vertebral III is hexagonal and longer than wide, with exception of GPIT/RE/09743 where it is wider than long. The lateral sides of vertebral III have equal lengths and are almost parallel to each other. The sulcus between vertebral III and pleural II is usually straight (87% of specimens), but sometimes convex (13%). Vertebral IV is hexagonal, usually wider than long, but almost twice as wide than long in GPIT/RE/09743 and longer than wide in GPIT/RE/09744 and GPIT/RE/09749. The lateral sides are generally of the same length, with exception of GPIT/RE/09738 and GPIT/RE/09743 where the posterior sides are shorter. The anterior margin of vertebral IV always crosses neural V, while the posterior margin generally crosses neural VIII, with exception of GPIT/RE/09738 and GPIT/RE/09747 (Fig. 4), where it overlaps the suture between neurals VII and VIII. Vertebral V is trapezoidal, more than two times wider than long and with convergent lateral margins that run across costal VIII. GPIT/RE/09747 (Fig. 4) has an anterior constriction in vertebral V that is unique to that specimen. A large contact between vertebral V and marginals XI is present in GPIT/RE/09743 and GPIT/RE/09747 (Figs. 4 and 11).

Pleural scutes. There are four pairs of pleural scutes in all specimens. Many specimens, adults and juveniles, show growth annuli on the lateral side of the pleurals (e.g., GPIT/RE/09744, GPIT/RE/09735, GPIT/RE/09741 and GPIT/RE/09743). Pleural I is the longest pleural and overlaps part of the nuchal, peripherals I–IV, and sometimes peripheral V (GPIT/RE/09738, Fig. 9), contacting directly marginal scutes I–IV as well as part of marginal V. The sulcus between pleural I and II is straight or sinuous, but without an anteromedial process, runs across costal II, and always contacts marginal V. Pleural II is rectangular, almost two times wider than long, and overlaps peripherals V–VII, directly contacting marginal VI and part of marginal scutes V and VII. The sulcus between pleural II and III is straight, without an anteromedial process, runs across costal IV, and, as documented by GPIT/RE/09743 (Fig. 11), contacts marginal VII. Pleural III is quadrate, wider than long, overlaps peripherals VII and VIII, and at least in GPIT/RE/09743, sometimes part of peripheral IX. It contacts partially marginal scutes VII and IX, and completely contacts marginal scute VIII. The sulcus between pleural III and IV runs across costal VI and contacts marginal IX, at least in GPIT/RE/09743. Pleural IV is the smallest and shortest of the pleural scutes, quadrate, and overlaps peripherals IX and X. It partially contacts marginal scutes IX and XI, and completely contacts marginal X. Pleural IV never contacts marginal XII. The sulcus between pleural IV and vertebral V contacts marginal XI, at least as once again documented by GPIT/RE/09743 (Fig. 11).

Marginal scutes. There are 12 pairs of marginal scutes in all specimens. The contacts with the pleurals are listed above. Marginals I–III are wider than long, do not form serrations, and are placed anteriorly on the carapace. Marginal IV is longer than wide, placed anterolaterally on the carapace, and does not overlap onto any costal bones. Marginals V–VIII are situated on the sides of the carapace, longer than wide, and do not overlap onto any costal bone. Marginals IX–XII are located at the back of the carapace and are wider than long or as wide as long, at least in GPIT/RE/09747 and GPIT/RE/09735 (Figs. 3 and 4).

Plastron. The plastron of Banhxeochelys trani has the typical testudinoid configuration composed of an entoplastron, and pairs of epi-, hyo-, hypo- and xiphiplastra. The anterior and posterior plastral lobes are about the same length and width. A well-developed bridge ranges from peripheral IV to VI connecting carapace and plastron. The anterior plastron margin is usually straight (14 out of 18 specimens), but sometimes concave (three out of 18), or anteriorly convex (GPIT/RE/09742 only). Most specimens lack a medial notch in the anterior plastral margin. At the margin of the contact between the gular and the humeral, a lateral inflection is present in 50% of specimens. The posterior plastron margin has a well-developed, triangular anal notch. Even the smallest specimens lack fontanelles.

Epiplastra. The epiplastra exhibit a thickened margin in visceral view that extends from the anterior margin until the mid-length of the epiplastra, followed by a posterior step, but not an overhang. No muscular insertion marks and posterolateral processes can be observed on the visceral view of the epiplastra (Figs. 3B, 6B and 8B). A pair of “ptychogasterid spikes” (i.e., anteriorly directly processes) is present in 78% of specimens at the anterior margin of the epiplastra (Fig. 6A).

Entoplastron. The entoplastron is centrally located between the epi- and hyoplastra, rhomboidal, and as long as wide. The anterior and posterior portions, as defined by the epi-hyoplastral suture, are about the same size in 73% of specimens. In other specimens it is either the anterior part larger (10%) or the posterior part is larger (10%). The entoplastron is always intersected by the gularohumeral sulcus anteriorly and by the humeropectoral sulcus posteriorly, either close to its posterior margin (in eight out of 12 specimens) or just at the posterior margin (in the remaining four specimens). The posterior process of the entoplastron is only visible in visceral view in GPIT/RE/09736.

Hyoplastra and hypoplastra. The hyo- and hypoplastra are about the same length and width, and longer than wide. The hyoplastra contacts the epiplastra and entoplastron anteriorly, suturally contact peripherals IV and V laterally, and the hypoplastra posteriorly. They are crossed anteriorly by the humeropectoral sulcus and posteriorly by the pectoroabdominal sulcus. The hypoplastra are crossed posteriorly by the abdominofemoral sulcus, suturally contact peripherals V and VI laterally, and the xiphiplastra posteriorly. The axillary buttresses originate on the posterior half of the hyoplastra, as seen on the right hyoplastron of GPIT/RE/09752, where a part of the axillary buttress is preserved. The inguinal buttress, on the other hand, originates at the center of the hypoplastra, as seen on the right hypoplastron of GPIT/RE/09736. Although this buttress is not fully preserved, its size (about 2.5 cm) and oval shape suggests that it was well-developed. No lateral keels were observed in any specimen on the hyo- and hypoplastra.

Xiphiplastra. Xiphiplastra are large, longer than wide, and crossed anteriorly by the femoroanal sulcus. A step along the lateral margins of the xiphiplastra is associated with the femoroanal sulcus. The visceral xiphiplastral lip is observed in GPIT/RE/09736, the only specimen where it is possible to see this character. The xiphiplastral lip is low, but wide, like the one in Rhinoclemmys, and extends from the most posterior end of the xiphiplastra (anal notch) to just posterior to the inguinal notch.

Plastral scutes. There are each one pair of gular, humeral, pectoral, abdominal, femoral and anal scutes. No inframarginal scutes (i.e., axillary and inguinal scutes) can be discerned. The gulars are longer than wide. The gulars are less than half as long as the median length of the plastral forelobe. The humerals have the shortest median contact of all scutes, are wider than long, and overlap the epiplastra, entoplastron and part of hyoplastra. The humeropectoral sulcus usually converges toward the posterior and either contacts or crosses the entoplastron. The intersection of the humeropectoral sulci on the entoplastron can produce a heart shape (Figs. 3 and 9). The pectorals are wider than long, almost as long as the gulars along their median contact, and placed entirely on the hyoplastra with exception of a minor overlap onto the posterior portions of the entoplastron in some specimens. The pectoroabdominal sulcus is usually straight along its full length, except in GPIT/RE/09754, which presents a long anterolateral notch. The abdominals have the greatest median contact of all scutes, are almost as wide as long, and overlap part of the hyoplastra posteriorly and more than half of the hypoplastra anteriorly. In GPIT/RE/09743 (Fig. 11), the left abdominal seems to overlap part of peripherals VI and VII as well. The femorals are wider than long and cover part of hypoplastra posteriorly and less than half of the xiphiplastra anteriorly. The femoroanal sulcus converges anteriorly and contacts the lateral margin of xiphiplastra along a small notch. The anals are large, generally longer than wide (with exception with GPIT/RE/09743, where they are as long as wide, Fig. 11), entirely placed on the xiphiplastra, and not fused with each other along the midline.

Phylogenetic Analysis

Matrix

Our morphological matrix is based on the recently assembled character taxon matrix of Garbin, Ascarrunz & Joyce (2018), which focuses on extant geoemydids and embraces polymorphic character observations. The matrix was modified through the addition of 16 new morphological characters (see Appendix S1 for descriptions) and 12 extinct, putative geoemydid species from the Eocene and Oligocene of the northern hemisphere: Banhxeochelys trani sp. nov., Bridgeremys pusilla (Hay, 1908), Echmatemys septaria (Cope, 1873), E. wyomingensis (Leidy, 1869), Guangdongemys pingi Claude et al. 2012, Hardella siamensis Claude, Suteethorn & Tong 2007, Isometremys lacuna Chow & Yeh, 1962, Mauremys thanhinensis Claude, Suteethorn & Tong 2007, Palaeochelys elongata (Gilmore 1931), Sharemys hemisphaerica Gilmore 1931, Sinohadrianus ezoensis Shikawa 1953, and Sinohadrianus sichuanensis Ping 1929. All fossils were scored based on descriptions and photographs available in the literature (Claude, Suteethorn & Tong, 2007; Claude et al., 2012), with exception of Banhxeochelys trani, and the North American geoemydids Bridgeremys pusilla, E. septaria and E. wyomingensis, which were scored based on first hand observations of relevant material. Following Garbin, Ascarrunz & Joyce (2018), the “polymorphic” method was chosen (i.e., 0 & 1; Campbell & Frost, 1993) for coding polymorphic morphological characters. For the list of morphological characters, specimens analyzed, and our character taxon matrix, refer to Appendixes S1–S3, respectively.

Our molecular matrix is also based on the one of Garbin, Ascarrunz & Joyce (2018), with three mitochondrial (12S, cytochrome c oxidase I, cytochrome b) and four nuclear loci (R35 intron, c-mos, Rag1 and Rag2) from the works of Honda et al. (2002), Spinks et al. (2004), and Le & McCord (2008). Aiming for a complete geoemydid phylogeny, the molecular data of some species (i.e., Batagur baska, Pangshura sylhetensis, Rhinoclemmys diademata, Cuora mccordi, Cuora yunnanensis, Melanochelys tricarinata, Mauremy japonica, Mauremys nigricans, and Geoemyda japonica) was added to the matrix, even though we did not have access to their morphological data. For details on sequence alignment and on how this matrix was produced, please refer to Garbin, Ascarrunz & Joyce (2018).

Analysis

We performed a total-evidence analysis (TEA) in Tree analysis based using New Technology (TNT) (Goloboff & Catalano, 2016) based on the morphological and molecular data. After we merged both matrices in TNT (Appendix S4), we performed the analysis with 5,000 replicates of random addition sequences, holding up to one million trees, followed by a round of Tree Bisection and Reconnection (TBR) branch-swapping from the trees held, and a Nelsen strict consensus. All characters were left with equal weight and morphological characters 3, 11–19, 22, 26, 30, 37, 38, 48, 49, 51, 52, 54–56, 58, 70, 79, 80, 88, 92 and 94 were run ordered (for details, see Appendix S1), following Garbin, Ascarrunz & Joyce (2018). The TEA yielded a total of 2,320 most parsimonious trees (MPTs) with 280 hits out of 5,000 replications (some replications overflowed) and a best score of 6,716. After the round of TBR, the same best score remained and the number of MPTs held went up to 133,736 trees (Appendix S6). The strict consensus of all these MPTs is shown in Fig. 13.

Figure 13 Strict consensus of 133,736 most parsimonious trees including all extant and fossil species on our matrix.

Fossil species are shown in bold. Major extant clades of Geoemydidae retrieved as monophyletic are marked in colors.

We then ran the IterPCR pruning command from TNT (Pol & Escapa, 2009) to identify rogue species that could be pruned from the trees in order to gain better resolution, followed by a strict consensus. This analysis suggested that all included fossil species should be pruned to gain resolution of 16 nodes, with exception of Guangdongemys pingi and Banhxeochelys trani. The reduced strict consensus with pruned species is shown in Fig. 14. The matrix was also run with a mild weighting factor of 12 (following Goloboff, Torres & Arias, 2018) and pruned after. The reduced strict consensus of this analysis is provided in Appendix S7.

Figure 14 Strict consensus of 133,736 MPTs after pruning all extinct species with exception of Guangdongemys pingi and Banhxeochelys train.

Major extant clades of Geoemydidae retrieved as monophyletic are marked in colors. Banhxeochelys trani is retrieved at the base of ingroup, Pan-Testuguria.

To investigate the phylogenetic position of each fossil species, we ran the strict consensus again, but this time, we included only one fossil at a time (excluding the other fossils from the consensus calculation, not from the matrix). The summary of the phylogenetic position of the fossils in each reduced consensus is given in Fig. 15. Sharemys hemisphaerica and the species of Sinohadrianus are not shown in this tree as the reduced consensus resulting from their individual analysis was not in agreement with that of other extinct geoemydids (Appendix S5).

Figure 15 Summary of the individual position of each fossil species in the strict consensus of 133,736 MPTs, keeping only one fossil species at a time.

This figure is based on the consensi from Appendix S5. Clades were reduced to genus name to minimize differences between consensi. Sharemys hemisphaerica and the species of Sinohadrianus are omitted here, as their consensus was not in agreement with that from other species.

For a final run, we ran the TEA on TNT with the same parameters described above (first paragraph, this session) excluding all fossil taxa with exception of Banhxeochelys trani. This analysis yielded eight MPTs with a best score of 6,667 and 1,743 hits out of 5,000 replications. The results were the same after the round of TBR branch-swapping. The reduced strict consensus of these eight MPTs is shown in Fig. 16. In this analysis Banhxeochelys trani had a different position by being placed as sister-taxon to crown Geoemydidae.

Figure 16 Strict consensus of eight most parsimonious trees resulting from analysis that includes Banhxeochelys trani as the only active fossil species.

This phylogenetic analysis followed the same parameters as the one for Fig. 13. Banhxeochelys trani is retrieved as sister to all extant geoemydid species. Major extant clades of Geoemydidae retrieved as monophyletic are marked in colors.

Discussion

Alpha taxonomy

As geoemydids appeared across the northern hemisphere in the early Eocene (Lapparent De Broin, 2001; Claude et al., 2012; Vlachos, 2018) in near synchrony, it is necessary to compare Banhxeochelys trani gen. et sp. nov. to putative Eocene/Oligocene geoemydids across the globe to establish its validity as a new species.

Geoemydids probably dispersed from Asia to North America during the Paleocene–Eocene Thermal Maximum (Lourenço et al., 2012), but their fossil record is mostly restricted to the Eocene. Although nearly twodozen species were named near the turn of the 19th–20th century (Hay, 1908), only nine Echmatemys species and Bridgeremys pusilla are currently recognized as valid (Vlachos, 2018). All known species of Echmatemys are characterized by the presence of extremely well-developed axillary and inguinal buttresses and hexagonal neurals (Vlachos, 2018) and can therefore be readily distinguished from Banhxeochelys trani, which possibly has well-developed inguinal buttress (see Description), though certainly not as well developed as those of Echmatemys. E. haydeni Leidy 1870b is notable for having octagonal neural IV, but can nevertheless be further differentiated from Banhxeochelys trani by the absence of keels and more elongate vertebrals.

Banhxeochelys trani differs from the North American Bridgeremys pusilla by having a larger adult size (Bridgeremys pusilla having a maximum plastron length of 16 cm; Hutchison, 2006) hexagonal anterolaterally short-sided neurals (only on the seventh and eighth neurals, most neurals are posterolateral in Bridgeremys pusilla); an octagonal fourth neural (Bridgeremys pusilla having an octagonal second or third neural); and a well-developed anal notch (poorly developed in Bridgeremys pusilla). In our personal observations of Bridgeremys pusilla (specimens analyzed in Appendix S2) we further observed a thick xiphiplastral lip and a ventral nuchal lip in this species, both absent in Banhxeochelys trani. In addition, Hutchison (2006) documents the possible presence of a hinge in Bridgeremys pusilla, which is also absent in Banhxeochelys trani.

Geoemydids appear in the early Eocene of Europe as well (Lapparent De Broin, 2001), but it is unclear if they dispersed from Asia or from North America (Joyce et al., 2016). About two dozen species have been named based on material from the Eocene of England, France, and Germany (Lapparent De Broin, 2001), but an alpha taxonomic revision of the group is still outstanding. We here agree with Hervet (2004a, 2004b) and Claude & Tong (2004) that two lineages are present in the Eocene, although we side with Claude & Tong (2004) by seeing less taxonomic diversity. The first lineage (“Palaeochelys sensu lato—Mauremys” of Hervet, 2004a; Palaeochelys of Claude & Tong, 2004) is best represented by rich material from Messel (Palaeoemys messeliana; Staesche, 1928) and Geiseltal (Borkenia germanica; Hummel, 1935), both in Germany, but their taxonomic status and variation are in need of revision. The second lineage (“Ptychogasteridae group” of Hervet, 2004b) is best represented by Geiselemys ptychogastroides (Hummel, 1935), also from Geiseltal, Germany. We mostly compare Banhxeochelys trani to these species, as they appear to be representative for the diversity of European geoemydids from the Eocene/Oligocene.

Palaeoemys messeliana (Francellia messeliana of Hervet, 2004a) differs from Banhxeochelys trani by having a small sized carapace (maximum 20 cm), a slight nuchal emargination, weak lateral keels, a lyre-shaped first vertebral with strong anterolateral constriction, hexagonal neurals with anterior short sides, a pygal bone intersected by the posterior margin of the fifth vertebral, and an entoplastron not intersected by the humeropectoral sulcus (Hervet, 2004a; Claude & Tong, 2004). Borkenia germanica differs from Banhxeochelys trani by having a weak nuchal notch (absent in Banhxeochelys trani), a hexagonal sixth neural with posterolateral short sides, absence of lateral keels, an entoplastron not overlapped by gular or pectoral scutes, a completely straight anterior plastron margin without any notch, and gular scutes wider than long (Hervet, 2004a). The differences seen in these two species appear to hold true for all other representatives of the Palaeoemys lineage.

Banhxeochelys trani differs from Geiselemys ptychogastroides by having less thick epiplastral lip, by lacking a posterior step, and shorter lateral spikes on the anterior plastral margin (“ptychogasterid spikes”; strong in ptychogastroides). Unlike Banhxeochelys trani, Geiselemys ptychogastroides has an octagonal neural II, and neurals III–VI that are hexagonal with posterior short sides, or sometimes rectangular. These two species have in common an entoplastron intersected posteriorly by the humeropectoral sulcus, a deep anal notch and a moderate xiphiplastral lip, short in thickness (seaming less thick in Banhxeochelys).

In Southeast Asia, two geoemydid species have been described from the Krabi basin (Chron C13R, Eocene–Oligocene boundary) in Thailand, Hardella siamensis and Mauremys thanhinensis (Claude, Suteethorn & Tong, 2007), proposed to be closely related to extant Hardella and Mauremys species. Two other geoemydid species, Guangdongemys pingi and Isometremys lacuna are known from the late Eocene Maoming basin of China (Claude et al., 2012), a locality 400 km away from the Na Duong coal mine. I. lacuna has been hypothesized to be more closely related to the old-world geoemydids (i.e., “three keeled Geoemydidae” of Claude et al., 2012), than to Echmatemys and Rhinoclemmys, while Guangdongemys pingi has been hypothesized to have a more basal position within the geoemydid crown clade (Claude et al., 2012).

Banhxeochelys trani differs from Mauremys thanhinensis by having a longer median keel that crosses neural I to suprapygal I, shorter lateral keels (crossing costals I–VII or restricted to costals III and IV), neurals with anterior short-sides (posterior in Mauremys thanhinensis), a first vertebral scute contacting only the first marginal, wide bridge peripherals, a median notch at the anterior plastral margin, and a thin xiphiplastral lip.

Unlike Hardella siamensis, Banhxeochelys trani has a longer median keel crossing neural I to suprapygal I, two lateral keels, an octagonal fourth neural, bridge marginal scutes that extend over the hyoplastron, an entoplastron intersected by the humeropectoral sulcus, and a deep triangular anal notch.

Banhxeochelys trani shows several similarities with Isometremys lacuna, like the presence of three carapacial keels, neurals II–V about the same size, wide vertebral scutes, a wider than long first vertebral scute, and an entoplastron intersected posteriorly by the humeropectoral sulcus. But unlike Banhxeochelys trani, I. lacuna has all neural bones with anterior short-sides and its median keel is located posteriorly only, crossing neural IV to suprapygal II (in Banhxeochelys trani it crosses all neurals).

As Guangdongemys pingi, Banhxeochelys trani has an octagonal fourth neural, the other neurals have anterior short-sides, the first neural is oval or rectangular, and the short costals II–V are about the same length. On the other hand, Guangdongemys pingi does not have carapacial keels, the entoplastron is not intersected by the humeropectoral sulcus, and vertebral I is narrower.

Many geoemydid species have been described from the Eocene/Oligocene of China, Kazakhstan, and Japan (Gilmore, 1931; Urata, 1968; Chkhikvadze, 1973; Claude & Tong, 2004), but as with the European geoemydid fauna, these species are in need of taxonomic revision (Danilov, 2018).

Of the 11 geoemydid species described from the Eocene–Oligocene of Kazakhstan (Chkhikvadze, 1970, 1971, 1973, 1990), we have chosen to compare Banhxeochelys trani with “Echmatemys” orlovi Chkhikvadze, 1970 and Zaisanemys borisovi Chkhikvadze, 1973 (not (sic!) “Echmatemys” borisovi Chkhikvadze, 1990) because the remaining species from Kazakhstan are described from small fragments (Danilov, 2018) and therefore do not provide a significant amount of information for comparison (Chkhikvadze, 1970, 1971, 1973, 1990).

Zaisanemys borisovi differs from Banhxeochelys trani by having narrower epiplastral lips that do not touch medially, a heart-shaped anterior plastral margin with a median notch, and an entoplastron not intersected by the humeropectoral sulcus. “Echmatemys” orlovi differs from Banhxeochelys trani by having a straight anterior plastral margin with strong lateral spikes (“ptychogasterid spikes”), a moderate to thick epiplastral lips (nine mm thick; Chkhikvadze, 1973) that do not meet at epiplastral midline, a moderate xiphiplastral lip, and the presence of axillary and inguinal scutes. As Banhxeochelys trani and the North American Echmatemys, E. orlovi has a deep anal notch (Chkhikvadze, 1973).

Three species are named from the Eocene/Oligocene of China: Palaeochelys elongata, Sharemys hemisphaerica, and Sinohadrianus sichuanensis. Palaeochelys elongata differs from Banhxeochelys trani by the following combination of characters: carapace with a median interrupted keel; variable shape of neurals, such as an octagonal third neural, first neural rectangular, second neural hexagonal with anterolateral short sides, and all other neurals with posterolateral short sides; small size, reaching a maximum carapace length of 24 cm; a great inflection on the margin of the gularohumeral sulcus; entoplastron not intersected by the humeropectoral sulcus; and a large axillary scute (Gilmore, 1931). Banhxeochelys trani also shares some similarities with this species, such as the presence of a deep anal notch and gular scutes that are longer than wide (Gilmore, 1931; Brinkman, 2008).

The other early Oligocene species from China, Sharemys hemisphaerica, differs from Banhxeochelys trani by the following combination of characters: very large sub-hemispherical carapace, no carapacial keels, a well-developed nuchal notch, only one suprapygal bone, pygal bone overlapped by the fifth vertebral, wider plastron (bridge to bridge width), anterior plastron margin with a median notch, a strong inflection lateral to the gular scutes, humeropectoral sulcus intersecting the entoplastron, anteriorly to the epi-hyoplastron suture, pectoroabdominal sulcus intersecting part of the hyo-hypoplastron suture, and a very short anal scute (Gilmore, 1931; Brinkman, 2008). Of all the species compared here, this is probably the most distinct species from Banhxeochelys trani.

Sinohadrianus sichuanensis from the middle Eocene has a similar neural series shape to that of Banhxeochelys trani, as both have all hexagonal anterior short-sided neurals, and an octagonal fourth and square fifth neural bone (Ping, 1929; Brinkman, 2008). However, Banhxeochelys trani is distinguished by having three carapacial keels (absent in Sinohadrianus sichuanensis), costal bones II–VIII with equal length on the inner and outer corners (costals V–VIII have slightly alternating lengths in Sinohadrianus sichuanensis), a short posterior plastral lobe, an entoplastron intersected by the humeropectoral sulcus, and a deeper anal notch.

The Japanese species Sinohadrianus ezoensis from the late Eocene of Hokkaido is known only from the inner cast of the carapace and the exterior part, where only the sulci outline is preserved. This species is distinguished from Banhxeochelys trani by having all neurals hexagonal with anterior short sides, narrower neural bones, shorter first costal bone, and the presence of two pygal bones: the first half-moon shaped and the second, square.

Of all above mentioned species, Banhxeochelys trani is most similar to the Maoming species Guangdongemys pingi and Isometremys lacuna. The Na Duong species shares with Guangdongemys pingi the presence of an octagonal fourth neural and remaining neurals hexagonal with anterior short sides, and with I. lacuna the presence of three carapacial keels and an entoplastron intersected by humeropectoral sulcus. It therefore is “intermediate” between these two Maoming species. However, Banhxeochelys trani does not have particular characteristics that makes it more similar to any Maoming species in particular, and is not phylogenetically closely related to neither Guangdongemys pingi nor I. lacuna (Figs. 14 and 15). Due to the unique set of characters that are present in this Na Duong material we herein conclude Banhxeochelys trani to be a new species.

Intraspecific variation

Intraspecific variation of morphological characters is either associated with gender (i.e., sexual dimorphism), space (i.e., geographic variation), maturity (i.e., ontogenetic variation), pathology (e.g., developmental malformations), phenotypic plasticity (e.g., the development of different morphotypes despite the same genetic basis), or regular genetic variation unrelated to any of the previously listed factors (Ridley, 2006). As paleontologists rely solely on osteological characters, a good understanding of intraspecific variation in skeletal morphology is important to this community, as variation based on genetic differences should be preferred when establishing new species or assessing phylogenetic relationships. Countless studies exist that summarize variation within extant turtle species (Sánchez-Villagra et al., 1995; Lovich et al., 1998; Garbin et al., 2016), but these typically focus on externally visible soft-tissue characters and therefore only have limited utility to paleontologists. Notably exception include Minx (1992) and Delfino, Fritz & Sánchez-Villagra (2010), which summarize variation in phalangeal formula in North American box turtles and trionychians, respectively, or Bever (2009a, 2009b), which detail variation and growth in the skull of Pseudemys texana and Sternotherus odoratus, respectively.

One reason why it is difficult to document skeletal variation in extant turtles is because well-prepared skeletal material is rare in museum collections (Garbin, Ascarrunz & Joyce, 2018). This is generally true for fossils as well, but particularly fossil rich localities or formations often yield large samples of turtles that can be used to document skeletal variation in extinct species. Large numbers of fossil geoemydids (i.e., more than 10 individuals) have previously been reported from the Eocene of Messel, Germany (Cadena, Joyce & Smith, 2018), the Eocene of Geiseltal, Germany (Hummel, 1935), and the Eocene of Wyoming (Gilmore, 1945; Brand et al., 2000), but no study has of yet properly summarized and discussed intraspecific variation based on this material.

We here are able to documented intraspecific variation of Banhxeochelys trani based on 30 near complete to complete shell specimens, representing 14 adults (i.e., midline length of hyoplastron greater than 70 mm), eight subadults (i.e., midline length of hyoplastron average of 60 mm), and eight juveniles (i.e., midline length of hyoplastron average of 40 mm; Table 1). As all individuals were collected from a single stratigraphic horizon and as the available variation cannot be organized into morphotypes, we here regard this collection to reflect variation found in a natural population of a single species.

Table 1 Measurement of shells of Banhxeochelys trani from specimens described in this study.

Specimen	Maturity	CML	VeIII ML	CoI MxL	NeIII MxL	PML	Ab ML	Hypo ML	
GPIT/RE/09731	AD	282	52	43	24	237	57	75	
GPIT/RE/09732	AD	>305		56	33			85	
GPIT/RE/09733	AD		55	49	29		65	78	
GPIT/RE/09734	AD			45	31	235		70	
GPIT/RE/09735	AD	288	55	53	28	260		78	
GPIT/RE/09736	AD					260		73	
GPIT/RE/09737	AD			56	33			79	
GPIT/RE/09738	AD	292	58	46		272	68	79	
GPIT/RE/09739	JV			26	14		34	38	
GPIT/RE/09740	JV		27	27				40	
GPIT/RE/09741	JV		28	22			40	43	
GPIT/RE/09742	JV			21	11			30	
GPIT/RE/09743	JV	170	30	30		155	37	42	
GPIT/RE/09744	JV		29	28	17			47	
GPIT/RE/09745	SB		39	43	23			61	
GPIT/RE/09746	SB			41	25			60	
GPIT/RE/09747	SB							56	
GPIT/RE/09748	SB			38	24			62	
Notes:

All measurements are reported in mm. Scute and bone nomenclature are in agreement with the text. ML stands for mid length; MxL stands for maximum length. CML is carapace mid length. PML is plastron mid length. Maturity stages are defined in the text; AD (adult), JV (juvenile) and SB (subadult).

Ab, abdominal scute; CoI, first costal bone; Hypo, hypoplastron; NeIII, third neural bone; VeIII, third vertebral scute.

Out of the 96 characters scored for this species in the phylogenetic analysis, around 40% show some degree of polymorphism. This variation is described in detail above (see Description above) and we just summarize some important characters here.

The most apparent variation we observe in Banhxeochelys trani pertains to the presence of lateral carapacial keels. In many extant geoemydid (e.g., Mauremys reevesii, Cyclemys dentata, Heosemys spinosa), juveniles have a three-keeled carapace, but the lateral keels are reduced or even lost completely in subadults and adults (Claude & Tong, 2004; Claude et al., 2012). This appears to be the case as well for Banhxeochelys trani, as juveniles and some subadults are tricarinate, while adults lack lateral keels.

We here also attribute changes in the presence of a notch in the anterior plastral margin to ontogenetic variation. In this study, a notch is present in 46% of juveniles and subadults, but only in 22% of adults. This indicates that a notch is commonly present in juveniles but probably gradually disappears during growth, perhaps due to the addition of bony material at the margin of the plastron. This appears to be the first time that ontogenetic variability is documented for this character.

The variation observed in the shape of the neural I (i.e., oval, rectangular, or triangular) is in agreement with that observed in other geoemydid species (Garbin, Ascarrunz & Joyce, 2018), as well as that observed in other cryptodires (Pritchard, 1988). Two specimens (i.e., GPIT/RE/09759 and GPIT/RE/09748) have a continuous series of hexagonal neurals with anterior short sides, which differs from the common condition of Banhxeochelys trani, where the series is pierced by an octagonal fourth neural and a square or rounded fifth neural.

Most specimens of Banhxeochelys trani have an entoplastron that is as long as wide, with anterior and posterior halves of about the same size. However, in three specimens (GPIT/RE/09754, GPIT/RE/09736 and GPIT/RE/09738) the anterior part of the entoplastron is larger, and in two other specimens (GPIT/RE/09752 and GPIT/RE/09755) the posterior part is larger. We conclude this to be random variation, as there does not seem to be a correlation between this variation and other, variable characters observed in these specimens. Although always intersected posteriorly by the humeropectoral sulcus, another variation observed in the entoplastron is the exact place of this intersection. In 67% of specimens, the sulcus crosses the most posterior suture of the entoplastron (in some way overlapping the ento-hyoplastron suture), and in 33% of specimens the intersection is between the epi-hyoplastron suture and the posterior suture of the entoplastron.

Most geoemydids, and testudinoids in general, can show sexual dimorphic characters in shell morphology (Ernst & Barbour, 1989). While males of terrestrial and semi-aquatic species usually have concave plastra, females tend to have flat ones (Pritchard, 1979). Carapace maximum length also varies between sexes, with females being up to three times longer than males in some species (e.g., Hardella thurjii; Pritchard, 1979). For other groups of turtles (like stem and crown pleurodires) other sexual dimorphic characteristics may apply, such as the presence of a more domed carapace and a narrower but broader anal notch in females (Ernst & Barbour, 1989; Sullivan & Joyce, 2017). The specimens in our sample of Banhxeochelys trani are either not sufficiently complete and/or show massive deformation, and we are therefore not able to determine the sex of specimens or distinguish between sexual related characteristics in adults. We finally also cannot comment on geographic variation of Banhxeochelys trani, as all specimens are from a single quarry.

Phylogenetic relationships

When the total-evidence matrix of 96 morphological characters and seven molecular loci is run with all fossil taxa deactivated, a fully resolved tree is retrieved, in which Geoemydidae is found as monophyletic relative to Testudinidae. When all fossils are activated, the strict consensus neither shows a clear phylogenetic position for Banhxeochelys trani nor any of the other putative extinct geoemydid species included (Fig. 13). This poorly resolved consensus furthermore only retrieves some extant clades as monophyletic, such as Cuora, Rhinoclemmys, and Heosemys, but not others, such as Pangshura and Batagur. After pruning all extinct species but Banhxeochelys trani and Guangdongemys pingi from the consensus, the resolution improves by 16 nodes (Fig. 14). In this reduced strict consensus, all main extant geoemydid clades are retrieved as monophyletic (e.g., Pangshura, Batagur, Mauremys, Cuora) and Guangdongemys pingi has a clear position as sister to Geoclemys hamiltonii, supported by a first vertebral scute that is longer than wide. Banhxeochelys trani, however, is still found in a polytomy at the base of Testuguria together with Malayemys, Orlitia, Siebenrockiella, and Testudinidae. The polytomy among extant testugurians is retained even when all fossils are pruned from the consensus (not shown). It is therefore clear that the inclusion of fossils negatively impacted resolution among extant taxa.

We here for the first time investigate the phylogenetic position of some putative Paleogene geoemydids in an explicit phylogenetic context. Until now, species had either never been phylogenetic investigated, or had been manually placed in molecular trees using hypothesized synapomorphies (i.e., Claude & Tong, 2004; Claude et al., 2012). As all fossils are placed in a basal polytomy in the total evidence analysis (Fig. 13), we here investigate their phylogenetic placement individually by pruning all fossils but one from a series of reduced consensus trees compiled from the total evidence analysis (summarized in Fig. 15). We discuss below the phylogenetic position of each fossil species in these reduced consensus trees (Appendix S5) and highlight morphological characters that support these placements. We do not assert this scenario to be the most correct hypothesis for geoemydid evolution, but as an additional step toward the comprehension of the evolutionary history of Testudinoidea.

In all reduced consensuses (Appendix S5), the ingroup clade (Testuguria) is supported by the following characters: presence of anterior and posterior musk duct foramina, pygal bone completely divided by the twelfth marginal sulcus, gular scute as long as wide (or wider than long), anterior region of the entoplastron larger than the posterior, and a deep anal notch. Again, these synapomorphies are not exclusive to all species. For example, both testudinid species Stigmochelys pardalis and Gopherus polyphemus, have a pygal bone that is not divided by the twelfth marginal sulcus, and some geoemydid species (e.g., Leucocephalon yuwonoi McCord, Iverson & Boeadi, 1995; R. areolata Duméril & Bibron, 1851) have an incompletely divided one.

The branch of Banhxeochelys trani is supported by the presence of lateral keels that are extending from the first to fourth pleural scute, an epiplastral lip that extends until the mid-length of the epiplastron or closer to the entoplastron, and no distinct processes at the hypo-xiphiplastral suture and at the epi-hyoplastron suture.

The position of Guangdongemys pingi as sister to Geoclemys hamiltonii is supported by a first vertebral scute that is longer than wide. The other Maoming species, Isometremys lacuna, is placed at the base of Testuguria and its branch is supported by the presence of three carapacial keels, a median keel placed posteriorly (along third, fourth and fifth vertebral scutes), posterior marginals VIII–XII that are not flared, and an entoplastron that is larger posteriorly.

Although the Krabi species Mauremys thanhinensis and Hardella siamensis present characters that could be attributed to extant genera inside crown clade Geoemydidae (e.g., contact between first vertebral and second marginal for Mauremys; and vertebral scutes that are as long as wide for Hardella; Claude, Suteethorn & Tong, 2007; Garbin, Ascarrunz & Joyce, 2018), both were recovered at the base of Testuguria, as sisters to Testudinidae, Geoemydinae, Batagurinae, and other fossil geoemydids. Autapomorphies of Mauremys thanhinensis include the presence of three carapacial keels, a posterior median keel, posterior marginals VIII–XII that are not flared, a fourth vertebral scute that is as long as wide, inguinal buttress insertion at fourth costal, humeropectoral sulcus intersecting the ento-hyoplastron suture, and subequal entoplastron areas (as defined by the epi-hyoplastron suture contact). The autapomorphies of Hardella siamensis are a posterior median keel, second pleural contacting the sixth marginal, a fourth vertebral scute as long as wide, no axillary and inguinal scutes, and a rectangular anal notch.

The North American species of Echmatemys and Bridgeremys pusilla were all placed along the base of Testuguria. Autapomorphies of Bridgeremys pusilla are a cervical scute notched at the posterior margin, a visceral nuchal lip, a first vertebral scute that is longer than wide, anterior musk duct foramina on the axillary buttress, posterior musk duct foramina on the eighth peripheral, a long epiplastral lip that almost reaches the epi-entoplastron suture, presence of a step posterior to the epiplastral lip, an entoplastron not intersected by the gularohumeral sulcus, and entoplastron anterior and posterior regions that are subequal. The amount of autapomorphies for Bridgeremys pusilla reflects the uniqueness of this small Eocene species, that was correctly removed from Echmatemys by Hutchison (2006). In a recent analysis, these taxa were retrieved in a more derived position near Mauremys (Vlachos, 2018), but it is difficult to compare results, as the data matrices used differ substantially in character and taxon sampling. In either case, both taxa are not associated with Rhinoclemmys, which implies the independent dispersal of geoemydids from Asia to North America.

Echmatemys wyomingensis is supported by the presence of a posterior median keel, posterior marginals that are not flared, a cervical scute notched posteriorly, anterior plastral margin without spikes, epiplastral lip reaching the mid-length of the epiplastron, and the presence of a step posterior to the epiplastral lip. E. septaria, by contrast, is supported by the absence of carapacial keels, flared posterior marginals, the presence of a visceral nuchal lip, sulcus between first and second pleural contacting sulcus between fourth and fifth marginals, pygal bone without a posterior notch, epiplastral lip reaching closer to the epi-entoplastron suture, the presence of a step posterior to the epiplastral lip, epi-hyoplastron suture without distinct processes, and an entoplastron with anterior and posterior regions about the same size. Interestingly, if the analysis is run including these two taxa as the only active fossils, they are not retrieved as monophyletic.

Palaeochelys elongata, as Guangdongemys pingi, is the only Paleogene species recovered as closely related to extant geoemydids (Fig. 15). The position of Palaeochelys elongata inside Geoemydinae is supported by third and fifth neurals with posterior short sides, an inguinal buttress inserted at the fifth costal bone, and absence of inguinal scutes (as opposed to the presence of this scute in Batagurinae). The Palaeochelys elongata branch is supported by an anterior plastral margin without lateral spikes (i.e., ptychogasterid spikes) and subequal anterior and posterior parts of the entoplastron.

Three extant geoemydids, Malayemys, Orlitia borneensis, and Siebenrockiella crassicollis were recovered along the base of Testuguria in all reduced consensuses, with exception of the reduced strict consensus trees that included Sharemys hemisphaerica or Sinohadrianus sichuanensis (Appendix S5; Figs. 8 and 10). In these trees, Malayemys is placed as sister to O. borneensis at the base of Batagurinae. Their placement there and as sister to each other is supported by molecular signal only. In the Sinohadrianus sichuanensis consensus, Siebenrockiella crassicollis is placed as sister to all Batagurinae, which is supported by many molecular characters as well as the presence of a fifth neural with anterior short sides. In the Sharemys hemisphaerica consensus (Appendix S5; Fig. 8), Siebenrockiella crassicollis is recovered along the base with other extant species.

As it is possible that the coding of some of the external fossil taxa included in this analysis includes errors resulting from various taphonomic processes (crushing, preparation, imaging), we ran a final analysis that activates Banhxeochelys trani as the only fossil taxon. In this analysis Banhxeochelys trani is retrieved as a pan-geoemydid species as sister to all extant geoemydids (Fig. 16). This placement is supported in all eight MPTs and in the reduced strict consensus (Fig. 16), by the presence of anterior and posterior musk duct foramina, a character that is usually considered to be a synapomorphy for (crown) Geoemydidae (Hirayama, 1985; Yasukawa, Hirayama & Hikida, 2001; Le & McCord, 2008). As other groups of testudinoids possess musk duct foramina as well, we agree with Garbin, Ascarrunz & Joyce (2018) that better sampling of basal taxa is needed to confidently clarify the use of this character in diagnosing geoemydids. This study at least recovers the presence of musk duct foramina as a synapomorphy of Pan-Geoemydidae (Banhxeochelys + crown group) and the presence of gular scutes that are wider than long instead, as a non-exclusive synapomorphy of crown Geoemydidae. This is consistent with the morphological diagnosis of Banhxeochelys trani, through which we could infer a priori that this species is a representative of Pan-Geoemydidae (see Diagnosis).

We are aware that the inclusion of fossils in a total evidence analysis, especially when most fossils have a lot of missing data, can add uncertainties to the matrix and cause distant branches to collapse, which could result in a broad basal polytomy. However, the large polytomy at the base of Testuguria does not imply that the phylogenetic position of the putative fossil geoemydids included in the analysis is not known at all. Instead, it is important to note that all reduced strict consensus trees include large clades of extant geoemydids that affirmatively do not include these fossils. We therefore can have high confidence that most of these Paleogene taxa indeed represent basal branching testugurians or geoemydids, not derived representatives of Palatochelydia or Geoemydinae. This, in return, is consistent with an early Paleogene divergence scenario as predicted by several molecular calibration analyses (Lourenço et al., 2012; Joyce et al., 2013; Pereira et al., 2017).

Paleoecology

The sample of more than 100 shells of Banhxeochelys trani described herein was collected from lacustrine lignitic shales (layer 80) in the Na Duong formation (Böhme et al., 2013). The layer that yielded this material is reconstructed to have been a large lake with anoxic bottom waters that was inhabited by a diverse fish and crocodile fauna (Böhme et al., 2011, 2013).

There are no signs that the specimens of Banhxeochelys trani studied herein were transported to the site where they were deposited, as the majority of specimens are articulated (in contrast to terrestrial mammals) and show no signs of mechanical erosion. We therefore interpret Banhxeochelys trani as an autochthonous, aquatic turtle. To a certain degree it may be possible to distinguish more terrestrial habitat preferences from more aquatic ones using skull shape (Claude et al., 2004), limb proportions (Joyce & Gauthier, 2004), or shell morphology (Pritchard, 1979), but as no skull or limbs are present and all shells are fully crushed, these sources of information cannot be used to further clarify the ecological habits of Banhxeochelys trani. We cannot speculate about dietary preferences, although it is notable that the Na Duong site has yielded an exceptionally diverse flora and fauna (Böhme et al., 2013) that certainly could support the full spectrum of dietary preferences observed in extant testudinoids.

Despite the great numbers of recovered shells, there is no evidence for gregarious behavior in Banhxeochelys trani. The fossils were excavated evenly distributed over an area of approximately 10,000 m2, at a rate of one individual per 100 m2. Only a single shell accumulation was found (fig. 14 in Böhme et al., 2011), which is composed of fish remains, a crocodile tooth, six geoemydids, and a trionychid shell. Crocodile bite marks on the shells led to the conclusion that this unique accumulation may represent a crocodile regurgitate (Böhme et al., 2013). Geoemydid turtles therefore represent an important food source for the Na Duong crocodiles.

Supplemental Information

Supplemental Information 1 Definition of morphological characters used for total evidence analysis.

Click here for additional data file.

Supplemental Information 2 List of museum specimens analysed in this study.

Click here for additional data file.

Supplemental Information 3 Mesquite file of the character/state matrix of morphological characters.

Click here for additional data file.

Supplemental Information 4 Character/state matrix of morphological and molecular (DNA) characters together used for the Total Evidence Analysis.

Click here for additional data file.

Supplemental Information 5 Phylogenetic trees for the analysis including only one fossil at a time.

Click here for additional data file.

Supplemental Information 6 All most parsimonious trees resulting from the Total Evidence Analysis.

Click here for additional data file.

Supplemental Information 7 Reduced consensus tree of the Total Evidence Analysis using implied weights method (K=12), after pruning Sinohadrianus ezoensis.

Fossils species are marked in bold lettres. Emydid species Malaclemys terrapin is marked with an asterisk to sinalize its position in the ingroup.

Click here for additional data file.

We thank our Vietnamese colleagues who facilitate the and participated in Na Duong paleontological expeditions of 2009, 2011 and 2012: Nguyễn Việt Hung, La Thễ Phúc, Đặng Ngọc Trần, Đồ Đức Quang, Phan Đồng Pha. We also thank Ingmar Werneburg for helping with access to material at the University of Tübingen and Henrik Stöhr for help with preparing specimens for transport to the University of Fribourg. We thank Eduardo Ascarrunz for his help with the phylogenetic analyses in TNT and Julien Claude, Igor Danilov, Ren Hirayama, and Márton Rabi for useful discussion. This manuscript was significantly improved though numerous constructive and thoughtful comments from Julien Claude and Evangelos Vlachos.

Additional Information and Declarations

Competing Interests

Author Contributions

Data Availability

New Species Registration

The authors declare that they have no competing interests.

Rafaella C. Garbin conceived and designed the experiments, performed the experiments, analyzed the data, prepared figures and/or tables, authored or reviewed drafts of the paper, approved the final draft.

Madelaine Böhme contributed reagents/materials/analysis tools, authored or reviewed drafts of the paper, approved the final draft.

Walter G. Joyce conceived and designed the experiments, performed the experiments, analyzed the data, prepared figures and/or tables, authored or reviewed drafts of the paper, approved the final draft.

The following information was supplied regarding data availability:

The complete total-evidence matrix using 96 morphological characters, three mitochondrial and four nuclear loci is available in Appendix S4. This data was used for the phylogenetic analysis.

The following information was supplied regarding the registration of a newly described species:

Publication LSID: urn:lsid:zoobank.org:pub:D2620202-9814-4F81-BFA4-043EB1B6F948.

Genus name: urn:lsid:zoobank.org:act:63270D91-AC3C-4883-8869-4F2B0DED6822.

Species name: urn:lsid:zoobank.org:act:4050AB6C-EEB6-4D01-A22D-B7285ECE2EA8.

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
