# Peer review of "A new testudinoid turtle from the middle to late Eocene of Vietnam"

_PeerJ, doi:10.7717/peerj.6280_

## Round 0.1 · original submission · Minor Revisions

Both reviewers suggested minor revision. Please address their commnents/suggestions point-by-point. This is an excellent paper, and we look forward to receiving the revised version.

·

Basic reporting

There are no issues with Basic reporting, the work is very good overall. Please see comments to the author, because I have structured my review in a different way

Experimental design

There are no major issues with Experimental Design. Please see specific comments to the author, because I have structured my review in a different way

Validity of the findings

There are no issues with the validity of the findings.

Additional comments

Thank you very much for inviting me to review this important manuscript. When I was invited, the message of the authors to me said that they believe I will enjoy reviewing this work. I think it was even better. It is so rare to receive a work on fossil turtles based on so many specimens, with so good preservation, and placed within a detailed stratigraphical context. Even if the authors only described 30 out of 100 shells, I am tempted to speculate that the new species described by Garbin, Boehme, and Joyce is among the best-known fossil turtle species in the world, ever. To my knowledge, only the testudinid Stylemys nebrascensis and the combined Echmatemys spp. from USA would compete the richness of the new taxon herein, but they would still stand far from the detailed and meticulous descriptions and figurations herein, and from the precise stratigraphical context of Banhxeochelys. This work is beyond doubt a landmark paper for the entire clade of turtles. I do not know if the authors agree with this view, but if so, please make sure to comment somewhere on how unique is the material you bring to our knowledge.

Another reason that gave me joy in reading this work, is because I know fully understand the reasons behind the latest work of Garbin, Ascarrunz, and Joyce (2018; ZJLS) on the polymorphism on Geoemydidae. With the addition of Banhxeochelys I am now able to see and understand that bigger and beautiful picture that Garbin, Joyce, et al have been looking over the last years.

The authors described the 30 most complete shells of this turtle, accompanied by high-quality illustrations, extensive comparisons with relevant taxa, and a new phylogenetic analysis based on a recently published matrix. Based on this analysis, the authors propose a new taxon; I have no doubt about that. Also, they present a lot of new and important information, not only for the new taxon, but for many other extant and extinct geoemydids as well.

I would absolutely recommend the publication of this work beyond any reasonable doubt, and I would urge the authors to continue on this line and describe the remaining 70 shells as well in the near future.

I have only some specific comments to make, either as some minor ones on the pdf, or some comments that require more explanation from my part (below); these proposed changes would not alter the ms significantly. In all cases, I hope that I was able to understand the intended meaning.

Perhaps the only part that I would urge to be expanded has to do with the large polytomy of their phylogeny. Half of the tree is not resolved, but this is not the problem. The problem is that I find little effort from the authors to understand and explain why is that. Besides the superficial comment on homoplasy and polymorphism (p. 6 of the pdf), the “dangerous” phrase (if taken out of context) of p.32 that “[…] is therefore clear that the inclusion of fossils negatively impacted resolution among extant taxa”, and an unnecessary effort of creating numerous reduced consensuses to improve resolution, I do not see any meaningful efforts to solve and explain this problem. There is no doubt that there is homoplasy and polymorphism in Geoemydidae, but how and why this is supposed to be related with our poor knowledge of phylogenetic relationships of this clade is beyond my understanding; especially when we have the tools to investigate this. Not to mention that numerous works after the landmark paper of Gauthier et al. (1988) proved the importance of fossils in resolving phylogenies of extant taxa…

In the points E and F below I try to provide my opinion on what the authors should do to address those issues properly. Briefly, I think that the authors should take a full advantage of the script IterPCR that they are already using (which actually proves that polymorphism is not issue, but conflict of characters and missing information are; see below) and should also try implied weights (which is a proved method to improve resolution and tackle homoplasy).

The authors claim in the title that the new taxon has some implications for geoemydid systematics. Well, I am not sure if it is clear which are they? It is a new species and most probably is a stem geoemydid, but these do not add any significant implications in geoemydid systematics.

Actually, the reader is left with the (wrong) impression that geoemydid phylogeny is a mess, polymorphism and homoplasy are to blame, morphology does not make sense, and that the inclusion of fossils in phylogenies causes chaos. I find it hard to believe that the authors actually would like to support these ideas and that these are the implications stated in the title (please check and revise if necessary). I had the same issues with the Garbin et al. 2018 (ZJLS) paper, and I am still seriously concerned in this ms as well.

Thank you again for giving me the opportunity to serve as a reviewer for this respected journal and especially for a ms of this quality. I sincerely hope that I helped the authors in the revision of their work.

General Comments

A| One species or more? Even if the authors say that this is a true population (which I agree), this does not necessarily support the idea of a single species. To the contrary, the identification of a true population or snapshot should start with the consideration if they indeed represent a single species or not. Even today, Vietnam is a hotspot of turtle biodiversity and of geoemydids in particular, and we have the actualistic information of the sympatry between several testudinoids in Vietnam and other places in SE Asia. So, if today we analyse a true population or a snapshot we will find more than one species of a genus, or subspecies of a species. So why not in your case? Is it possible that the variation you record could be explained by the existence of more than one species? How is the polymorphy distributed in the specimens? For example, most of the polymorphies are scored (as far as I can tell from the figures and text information) on the specimen GPIT/RE/09743; does this specimen has a distinct morphotype compared to the rest? Or it is because it is a juvenile (so its morphology should not be taken into account for resolving the relationships of the species; see D3 below). What do the other 70 specimens tell you?

I have no doubt that Ba.trani is a new species, but I naturally wonder what if you have more than one species. Several of the characters you find as polymorphic, I used them to define different morphotypes in Echmatemys and to attribute them to previously named taxa. With this concept in mind (and if you have some nice distribution of the morphotypes of neural shapes and/or hu-pec sulcus vs. the entoplastron) you might have up to 3 different species at least. I do not know how you or others would react on that, but I personally think that is more likely to have more than one species than to have ONLY ONE. As 70 specimens are yet to be described, I believe that the best approach would be to name only one species, but to distinguish a set of morphotypes of Ba.trani which could be used as working hypotheses for a future distinction of taxa (in species/subspecies level) within your assemblage. I firmly believe that your assemblage is one of the few, or the only one, where such a work could be made. I am working on a similar case (but with much fewer shells), wherein a “snapshot” or a “true population” of 15 individuals there are beyond any doubt two different genera and up to 3 different geoemydid species. I think that you have a similar case, with at least two different clades of turtles, and more than one geoemydid species.

In any case, I would definitely welcome a comment on that.

B| After reading the ms, I am not sure if the authors consider the new taxon as a stem geoemydid or a testugurian. In the systematic list, it is listed under Pan-Testuguria, in the diagnosis they say is a member of Pan-Geoemydidae, in the final tree is in the stem of Geoemydidae. I would say that there has to be a bit more consistency on that. If the final position of the taxon is not certain, then I think that the addition of a final taxonomic conclusion would definitely help the reader. I, personally, believe that the new taxon is a stem geoemydid, and its recovery in this basal testugurian polytomy is not its problem, but actually the inherent problematic position of Testudinidae from the reference Garbin et al. (2018) phylogeny (see other relevant comments I made herein).

C| Diagnosis. After seeing the diagnosis and the comparisons, I would like to suggest considering the utility of the following characters as part of the combined diagnostic characters of trani: the wide, trapezoid, cervical scute and the wide suprapygal 2. These are certainly different from the Chinese “similar” taxa.

D| Matrix/Phylogenetic analysis: I have checked the matrix overall and the scorings of the new taxon in detail.

1| There are some inconsistencies between the text file and the tnt file. Some characters that are scored as inapplicable in the word file appear as “?” in the tnt file. Please check and revise if necessary.

2| If I understand correct, there are some extant species with DNA data but no morphological data. If so, please explain this in detail in the ms (i.e. which are those taxa).

3| If I understand correct, some of the polymorphisms scored by the authors are the result of scoring both the adult and juvenile morphologies (e.g., presence, number, and position of keels) and/or scoring the anomalies as well (e.g., the number of neurals; the number of vertebrals). I would like to point out that I do not think that is a correct practice scoring those cases, as neither the juvenile nor the anomalous morphologies should be considered as useful to add anything to the solution of the phylogenetic relationships of the taxon. In fact, I am certain that scoring those cases can only add problems. The phyl. analysis of the new taxon should be focused, in my opinion, only on the adult non-anomalous specimens (considering that the authors have the privilege of working with 100 fossils, excluding some of them shouldn’t be any problem). Of course, those cases should be mentioned in the text (as the authors have done already), but simply they should not be scored and taken into account for the phyl. analysis (in that case a note should be added in the text).

4| I have some comments regarding the scorings of some specific characters on the new taxon:

Ch8: flared posterior marginals> it should be “peripherals”, right? If so, please correct it to the text as well. The new taxon is scored as 0/1, but no such information is given in the text. Please check and revise

Ch12: number of suprapygals. Scored as 0/1, whereas in the text is mentioned that all specimens have only two suprapygals [l.195]

Ch26: the new taxon is scored as 0/1, but you mention that the sixth vertebral in the single specimen that shows this morphology is an anomaly. Are you certain you should score this anomaly as a polymorphy of the taxon?

Ch27: I am not sure about some of the scorings of this character. In two of the specimens where the notch is scored as present (09735, 09738) there is some deformation involved as well on one side of the vertebral I/cervical/marginal I association. Any why 09733 is not scored as having the notch as well?

Ch29: This character is scored as polymorphic (implying that there are at least two specimens where the visceral view of the nuchal can be observed and scored). However in the text is mentioned “The ventral side of the nuchal is exposed only in GPIT/RE/09751, but it is not possible to see any characteristics due to bad preservation”. So I do not understand how this character is scored. I mean, if you truly have not observed this character it should be scored as ?; as this character has only two states, scoring as ? or 0/1 would be the same for the analysis, but it is like you imply polymorphism where actually there is missing information. Please check and revise.

6| Uploaded files: Please consider uploading the Mesquite file with the character/state names of the morphological matrix separately. Please replace the terminal taxon “testu” with the genus and species name of the new taxon herein for overall consistency. Please upload the ctf file of the MPTs of the full run. Please upload the iterPCR results, especially the part of the conflict of characters, and missing information.

E| IterPCR. I would like to kindly ask to clarify if you run the actual script or the built-in command in TNT. It seems to me that what you actually did was only the first part of the IterPCR, which is not different than the pruning taxa command; it seems that you did not complete the character/taxa evaluation which is actually the juicy part of the method. I find it hard to believe that you have been able to run the full IterPCR method on the 700.000 trees from a computational point of view. I strongly suggest running the full IterPCR on a subset of trees which reflect the topology (e.g., I selected the first 1000 trees and is the same topology more or less) and try to find out why exactly you have this polytomy.

When you run the script, the reasons of conflict are clear (please SEE FIGURE inserted in the first page of the annotated pdf). The script highlights the conflicting characters and the missing characters which if scored could help to resolve the polytomies. First of all, polymorphism is not really an issue (see cells marked in red). Only few characters scored as polymorphic in a handful of taxa are returned as conflicting. Also, the multiple positions of the new taxon (the most polymorphic of the extinct taxa) are not the result of polymorphism. trani has only one conflicting character which is not polymorphic. The main problems are the presence of several conflicting (not polymorphic) characters (marked with green) and the amount of missing information which could be useful to solve the polytomies (marked with blue). Especially half of the extinct taxa contain a large amount of missing information, which appears to be crucial for resolving their position. Also, the script analyses the jumping positions of some clades (e.g., Rhinoclemmys and Heosemys) as the result of conflicting characters; although many of these conflicting characters in the included species are polymorphic, their reconstructions on the node of these clades are rarely polymorphic.

As the problem of the missing characters in the extinct taxa appears to be a serious one, I would reconsider the necessity of adding most of the extinct taxa in this phylogenetic analysis (they should remain in the comparisons, however). In fact, our knowledge about their position has not improved after this work. We thought that they are pan-geoemydids, they are shown as pan-geoemydids or pan-testugurians herein. Why adding them in the analysis if they only create noise?

F| I am not sure if I understand why the authors went after all this trouble with the various consensuses with the various pruned taxa; also the authors notified us during the review that the new taxon was selected as not to be pruned (otherwise it would be pruned as well). I am not sure either if all these trees are necessary to be presented. I understand that this is a big polytomy, and is OK. This is the result of the current matrix. The simplest way would be that the authors present the strict consensus of the full analysis as a supplementary information, and then the most stable reduced consensus tree (even if the new taxon is not there…); it is, however, important to mark on the reduced strict consensus the various positions of the pruned taxa. Also, the authors should use the term “reduced strict consensus” when they present trees where pruned taxa are not shown, and “strict consensus” when all analysed taxa are shown.

Also, you need to be clear any time you mention characters or synapomorphies in which set of trees are calculated. If I understand correctly, they are mostly calculated on each reduced consensus tree. For that reason, it is kind of difficult to follow and even to interpret those “synapomorphies” in a meaningful way.

I would recommend trying some analyses under implied weights, which are shown to outperform other methods in terms of retrieving correct groups (see the recent paper of Goloboff et al. 2018 in Cladistics). Maybe implied weights could help to improve the resolution of your trees.

If implied weights do not work then I suggest the following:

as the full analysis does not work, I suggest that the authors focus first on resolving the position of the new taxon. In this sense, the last tree that the authors present is fine, and I am happy about that. Ba trani is a stem geoemydid.
Then they could move on and investigate the position of the other taxa in their new matrix. Now, when all taxa are included, Ba.trani collapses to Pan-Testuguria, and the authors should try to explain why (see iterPCR). I believe that this is because of the inherent problems of the matrix regarding Testudinidae (numerous characters that could resolve the position of Testudinidae are not included, especially from the cranium, because this is shell-only, geoemydid matrix). If so, actually the inclusion of Ba.trani (a stem geoemydid) in the matrix is responsible for resolving the polytomy of Garbin et al (2018)!

·

Basic reporting

It is a good paper, clearly written. The literature is fine, figures and tables are good, I also appreciated the supplementary material (useful and well organized). Overall, I agree with the conclusions (see remarks for phyl. analysis and diagnosis however).

Experimental design

as it is organized, the paper is a clear taxonomic and systematic account. no problem.

supl mat. no 2 should be updated, the list of specimens studied is not complete (even if things were scored from the literature we should know which specimens or plates were used).

Validity of the findings

things are fine, just see my remark for the diagnosis that can certainly be improved, the results of the phylogenetic reconstruction could be artefactual and could be discussed a bit more (high degree of polymorphism in the fossil studied).

Additional comments

This paper describes a new genus of turtle from the Paleogene of Vietnam. The fossil record of turtles in South East Asia is not well known and this description adds up an interesting new account to what is currently known.

The description is based on a very large amount of material that has several similarities with something I described some years ago in Southern China (not too far from Loc Binh discrict actually (about 300/400 km).

It is a good paper.

Some remarks/comments that may help you to improve the ms:

1. The lack of phylogenetic relationships between Guangdongemys (or other taxa) and Banhxeochelys two may be real or rather an artefact of the morphological characters selected and total evidence (I know you have done your best: but many of them are polymorphic in several taxa, and may make branches collapsing early in the tree...this is often the case when using total evidence methods and include fossils with polymorphic character: mostly because this polymorphic characters, as they are treated adds uncertainties; this makes distant branches collapsing and the tree looking like a long basal polytomy until molecular characters do their job). If you look at the number of character diagnosing the fossil from south china and your fossils: they are very few (and we also know that these characters can change within a single genus). I would certainly nuance even more the result of the phylogenetic analysis; even if the provision of a matrix of character was very valuable in supplementary material. Actually, I am afraid that other scientists may consider that approach as a solution for justifying easily the erection of new taxa. Note that your fossil is one of the most variable, this can explain the position that was optimised by phyl. reconstruction methods.

2. In that section and in suppl material S2, you should clarify what you consider as belonging to Palaeochelys. the specimen list I got in suppl material is incomplete and it seems the document you uploaded was a previous version (at the end of the author list there is a XXXX).

3. Intro:
I would certainly avoid to say that Palaeoemys and ptychogaster are "European lineages". The western european fossil has long been regarded as evolving independently from Asia but we have no solid ground for that (the fossil record in Asia is too weakly known...and just by looking at your results, we can speculate it is not the case ).

4. Age of Krabi is very late Eocene or very early Oligocene (Chron C13R), this is the more precise approximation in the literature so far.

5. Diagnosis should be reworked to make sure it is not ambiguous..
In the diagnosis, it is said that the taxon is assigned to testudinoidea becuase of the contact between plastral and marginal scute. This is not a clear apomorphy of crown testudinoids, since the few work concerning phylogeny of testudinoids including cretaceous and paleogene material shows that the contact between plastral and marginal scute could have occured independently in testuguria and emydidae (see Tong et al., Wutuchelys paper 2017). Furthermore, other turtle families are known to have lost these scutes (eg., pleurodires). The presence of cervical scute should be removed from the diagnosis.

While the shape of neural 4 and 5 was interesting as a diagnostic feature, it seems that at least two specimens in the referred material did not match that (at least 10 %). I would therefore rewrite the part dealing with neurals, rather writing that neural 4 or 5 can be either octogonal and squared respectively or hexagonal. But doing that, the diagnosis fails to differentiate clearly the taxon from Mauremys (and also maybe Sacalia, and some species in the "Palaeochelys - Mauremys" waste basket of Hervet). It would be important to further differentiate Banhxeochelys from these in the diagnosis. The rather narrow and hexagnoal vertebrals (V2-V3) in adults could be something to add in the diagnosis which further underlines its resemblance with Guangdongemys.

In the same respect, it seems from figures that sometimes the entoplastron is not interesected by the HP sulcus (there is some variation here too; see line 305 of your description). Maybe you should offer a longer diagnosis with elements of comparison for taxa that could be close (Mauremys, Palaeochelys, ?P elongata, Isometremys, Guangdongemys). You do that well at the end of the MS, but a diagnosis should be the primary useful part of the taxonomic account; so I think you can improve it.

6. l. 129. I do not understand why the genus is abbreviated with the two first letters (Ba. rather than B.)

7. If this is available, it would be good to provide a better view of the epiplastral lips (visceral view of the anterior plastron) -fig 8 B is the only illustration for that, and it is not super clear- and also possibly the extent of these lips on xiphi and hypoplastron. You are not discussing the indentation on the lip (something present sometime in other species), you are also not telling whether the not so well defined "ptychogaster" pike of Hervet is present or not.

8. It is a pity that the buttress can not be observed, does that mean that they were extremely reduced ? Maybe isolated plates could help ? The bridge looks rather short and inguinal and axillary notches are very narrow, is it actually the case ? Do we know any appendicular material, vertebra that could be referred to that species or to testudinoid indet ?

9. This is not exactly the scope of the paper, but I really wonder why there is no carettochelyids in this locality. Maybe it could help to refine the age and suggest that you are rather late in the Eocene (probably later than Maoming) or taphonomic/paleoecological reasons. We have no trionychid in Krabi as well (and hundreds of shell were found).

julien CLAUDE
Institut des sciences de l'evolution
Université de Montpellier

---

## Round 0.2 · accepted · Accept

Thank you for addressing the reviewers' comments in detail. One minor correction is needed: It should be "middle" and "late" Eocene per ICS.

I recommend the revised version of the manuscript for acceptance for publication.

#